# Preparation and Antimicrobial Characterization of Poly(butylene adipate-*co*-terephthalate)/Kaolin Clay Biocomposites

**DOI:** 10.3390/polym15071710

**Published:** 2023-03-29

**Authors:** Raja Venkatesan, Krishnapandi Alagumalai, Seong-Cheol Kim

**Affiliations:** School of Chemical Engineering, Yeungnam University, Gyeongsan 38541, Republic of Korea

**Keywords:** poly(butylene adipate-co-terephthalate) (PBAT), kaolin, microstructure, food packaging

## Abstract

The biodegradable polymer poly(butylene adipate-co-terephthalate) (PBAT) starts decomposing at room temperature. Kaolin clay (KO) was dispersed and blended into PBAT composites using a solution-casting method. Fourier-transform infrared spectroscopy (FTIR), X-ray diffraction (XRD), scanning electron microscopy (SEM), and transmission electron microscopy (TEM) were used to evaluate the structure and morphology of the composite materials. PBAT/kaolin clay composites were studied by thermogravimetric analysis (TGA). The PBAT composite loaded with 5.0 wt% kaolin clay shows the best characteristics. The biocomposites of PBAT/kaolin [PBC-5.0 (37.6MPa)] have a good tensile strength when compared to virgin PBAT (18.3MPa). The oxygen transmission rate (OTR), with ranges from 1080.2 to 311.7 (cc/m^2^/day), leads the KO content. By including 5.0 wt% kaolin 43.5 (g/m^2^/day), the water vapor transmission rate (WVTR) of the PBAT/kaolin composites was decreased. The pure PBAT must have a WVTR of 152.4 (g/m^2^/day). Gram-positive (*S. aureus*) and Gram-negative (*E. coli*) food-borne bacteria are significantly more resistant to the antimicrobial property of composites. The results show that PBAT/kaolin composites have great potential as food packaging materials due to their ability to decrease the growth of bacteria and improve the shelf life of packaged foods.

## 1. Introduction

Thin-layer materials called biopolymer films and coatings have been used for a long time as food wrappers, carriers for transferring food products, and packing materials to preserve food products [1,2]. They perform as an excellent barrier against the transmission of oxygen and water vapor to prevent the undesired mass transfer and deterioration of food, thereby extending their shelf life and increasing food quality [3,4,5]. Biodegradable composites based on bioplastics are commonly used to produce food containers, storage carriages, and packing for food products [6,7]. These entirely prevent the circulation of O_2_ and H_2_O vapor, decreasing unwanted mass transfer and food spoilage to extend shelf life [8].

Biodegradable materials have received a lot of focus in recent times [9]. Because of its multiple benefits, such as being fully biodegradable, thermally stable [10], and flexible [11], for potential uses in food packaging, PBAT is a viable choice. Moreover, with its capacity to be produced through compression molding, it may be used as a material for food packaging [12]. However, to fulfill the specifications for its use in food packaging, PBAT’s present antimicrobial activity is poor. As a result, it is still important to enhance the relevant characteristics [13,14,15]. Biodegradable polymer materials have been investigated as a desirable replacement for petroleum-based polymers to fulfill food packaging demands because of their characteristics of almost unlimited feedstock, biodegradability, and reduced costs [16,17,18]. PBAT has been chosen over all other biopolymers for its distinctive characteristics of being commonly available and thermally stable [19,20,21]. However, PBAT biopolymers are a difficult alternative for food packaging materials due to their poor mechanical properties and limited capacity for forming an efficient oxygen and water barrier. Moreover, PBAT’s poor antibacterial characteristics prohibit its application in food packaging [22,23,24]. Therefore, PBAT’s main priority still is to increase mechanical properties and water and oxygen barrier characteristics while maintaining the desired antimicrobial activities.

To enhance the mechanical properties, barrier, and antimicrobial activities of biopolymers, fillers such as metal oxides and montmorillonite have been introduced [25,26,27,28]. It is believed that kaolin clays have a high potential for increasing biopolymer composites. Well-known fillers such as clay particles are often used in polymer composites due to their favorable characteristics [29]. They are used in several industrial applications, including biomedical, tissue engineering, and food packaging, as a result of their improved properties [30]. The most common clays are phyllosilicates, namely, layered silicates such as montmorillonite, talc, and kaolin [31,32]. Kaolin is chemically formed by two layers: octahedral AlO_2_(OH)_4_ and tetrahedral SiO_4_ [33]. Furthermore, thermoplastic amylose/kaolin composites exhibited mechanical and thermal stability [34]. Recently, the increase in the mechanical and barrier characteristics of semolina film by adding kaolin has led to the use of these composites in packaged foods [35]. The research on PBAT/kaolin composites, therefore, is rarely addressed when considering materials for packaged foods.

In this paper, solution casting of PBAT and kaolin was used to produce several kinds of PBAT/kaolin composites with different concentrations. Characterizations, such as Fourier-transform infrared spectoscopy (FTIR), X-ray diffraction (XRD), transmission electron microscopy (TEM), scanning electron microscopy (SEM), and mechanical tests, were then used to evaluate the physiochemical properties and tensile strength of the composite materials. Kaolin was introduced to PBAT composite materials as an antimicrobial agent and for testing purposes against E. coli and S. aureus, two food-pathogenic bacteria. Finally, we measured the rate of water and oxygen vapor transmission in PBAT and PBAT/kaolin composites in an attempt to further study the barrier properties. The results provided an opportunity for the more extensive use of PBAT/kaolin clay composites in food packaging by demonstrating that kaolin clays increase mechanical, thermal, and antimicrobial activity while lowering barrier characteristics.

## 2. Materials and Methods

### 2.1. Materials

PBAT with a molecular weight (Mw) of 14.2 × 10^4^ g mol^−1^ and a melting temperature of 110–120 °C was received from M/s BASF Ltd. in Tokyo, Japan. Samchun Chemicals provided the kaolin (marked as KO) clay for this study, with an average distribution particle size of 4 µm. Table 1 shows the chemical composition of KO, including CaO, K_2_O, and Fe_2_O_3,_ acting as the major impurities. Analysis reveals that silica (SiO_2_: 47.85%) and aluminum oxide (Al_2_O_3_: 37.60%) were the major compositions. All chemicals were used from their source without any further purification.

### 2.2. Preparation of PBAT/Kaolin Clay Composites

The preparation of PBAT/kaolin composites via solution mixing and drop casting [36] is shown in Figure 1. In 100 mL of chloroform, 2.0 g of PBAT polymer pellets were dissolved over a period of constant stirring to form a clear solution. The (1.0, 2.0, 3.0, and 5.0 wt%) KO materials dispersed immediately in the solution. The solution was placed into a Petri dish and then dried for 48 h at 40 °C in an oven [37,38]. The final stage was to dry the prepared films under a vacuum for 48 h to remove all residual solvents. After the majority of solvents had evaporated, the PBAT/kaolin composites were formed. After the film was dried for 8 h at 60 °C in a vacuum to become clear of residual solvents, the Petri dish was then removed to show the dried composite materials. The ratios are, respectively, PBC-0.0, PBC-1.0, PBC-2.0, PBC-3.0, and PBC-5.0. PBAT/kaolin composite films with wt% of 0.0, 1.0, 2.0, 3.0, and 5.0 of kaolin are summarized in Table 2. Films with a uniform thickness of 0.05 to 0.1 mm were developed. The obtained specimens were separated into 2.0 × 5.0 cm pieces for tensile strength.

### 2.3. Characterization

#### 2.3.1. Structural Characterization

With the help of a Fourier-transform infrared (FTIR) spectrophotometer (PerkinElmer, Spectrum 100, Waltham, MA, USA) with a resolution of 4 cm^−1^, the attenuated total reflectance Fourier-transform infrared (ATR-FTIR) spectra of the PBAT composites were studied. The composite film was matched and placed directly on the ray-exposing plate. Around 400 and 4000 cm^−1^ of wavelength was utilized to record the spectra. The characterization of the PBAT composites was performed using an analytical X-ray diffraction meter (Rigaku) at room temperature (PANalytical). CuKα served as the radiation source (λ = 0.15406 nm), and rectangular sections of each film, approximately 2 × 2 cm, were placed on a glass slide to fabricate samples.

#### 2.3.2. Morphological Studies

The morphology and structure of the PBAT composites were studied with scanning electron microscopy (SEM) (Hitachi, S-4800). The 10 nm gold–palladium was sputtered onto dried film samples using a Leica EM ACE200 sputter coater after being attached to a metal stub using double-sided carbon tape. Images were captured at an accelerating voltage of 20 kV. Transmission electron microscope (TEM) images were recorded with a JEM-2100 (JEOL, JEM-2100, Japan) at an acceleration voltage of 300 kV. Direct casting from the solution of the imaging samples onto copper grids followed by room-temperature drying has been used.

#### 2.3.3. Thermogravimetric Analysis (TGA)

The thermal stability evaluation of PBAT composites is performed using TA instruments (QA 50), and the curves are of samples analyzed using the Universal V4.5A program. The weight of each sample is around 5.5 mg, and the temperature range for the measurement is about 40.0–700.0 °C with increments of 10 °C/minute at the rate of 10 °C min^−1^ under nitrogen atmosphere (50 mL min^−1^).

#### 2.3.4. Mechanical Properties

Instron material testing equipment (Instron 5545, MA) with a 1 kN load cell was used to evaluate the strength of composites in terms of tensile load. According to (ASTM D638-14) ASTM standards [39], the samples were examined at a cross-head speed of 10 mm/min with gauge lengths of 25 mm and dimensions of 150 × 25 × 0.04 mm. The gauge length and thickness values for each sample were noted. The cross-head speed was held constant at 1 mm/s for testing the specimens. At 50 ± 5% RH and 23 ± 2 °C, the tests were conducted. We collected and evaluated strength and distance to look for relationships between stress and strain. Then, average values for each material's tensile strength were calculated for each sample. All items underwent 3 independent tests. The values of the tensile strength were given in MPa.

#### 2.3.5. Barrier Properties

The oxygen transmission rate (OTR) measurements for PBAT/kaolin composites were studied using a Noselab (ATS, Concorezzo, Italy) in accordance with the ASTM D3985-17 standard procedure [40]. Five different testing locations were utilized to evaluate the composite material, and the average result was taken. Whenever the material was at room temperature, preparation occurred. To use a Lyssy L80-5000, the water vapor transmission rate (WVTR) of the PBAT and PBAT/kaolin composites was evaluated in accordance with ASTM F1249-90 [41] at 100% RH and 23 C. The examination was performed 5 times, and an average result was calculated.

#### 2.3.6. Measurements of the Contact Angle

The wettability of PBAT and its composite films was tested using a contact angle analyzer [35]. A small portion of the film was placed on a solid support with a flat base to be evaluated. The sessile drop technique was used to conduct the measurements in ambient conditions. Images were captured every 20 s using a high-resolution CCD camera after a water droplet (~2 µL) was thrown onto the specimens. With a contact angle meter from DataPhysics Instruments, OCA-20 Korea, the surface water contact angles were measured at room temperature and used to determine the hydrophilic nature of PBAT and its composites.

#### 2.3.7. Antimicrobial Property Measurements

The zone inhibition method was employed to investigate the antimicrobial property of PBAT and its composites [42]. Briefly, a specific amount of bacteria culture (0.1 mL) was added to nutrient agar plates. The bacterial zones of inhibition were measured by covering the bacteria colonies with a circle film of 5 mm diameter and incubating them at 37 °C for 24 h.

#### 2.3.8. Statistical Analysis

The statistical significance of every result was evaluated with ANOVA in SPSS 21 (IBM, New York, NY, USA). The data are provided as mean and standard deviation. A one-way analysis of variance was employed to determine statistical differences, and a result of *p* ˂ 0.05 was the value of the maximum.

## 3. Results and Discussion

### 3.1. Characterization of Structural and Morphological Analysis

The morphological and structural properties of the PBAT and PBAT/kaolin composites are presented in Figure 2a. The vibration Si-O, which represents the absorption of kaolinite materials, appears in the absorption bands that appear at the wavelength range of 1010 and 1022 cm^−1^. The peak close to 3000 cm^−1^ is related to C-H stretching for the aliphatic and aromatic regions. With an adjacent CH_2_ peak at 720 cm^−1^, C-O in the ester bond showed a strong peak at 1710 cm^−1^. Figure 2b shows the XRD patterns of PBAT/kaolin composites at different weight percentages. The maximum values of 19.15°, 20.80°, 26.55°, 28.99°, and 50.07° are observed.

These results, which are in good accordance with the results of additional studies and could be suggestive of the semi-crystalline structure of kaolin [43], are also in line with previous studies. The composite PBAT/kaolin clay film with kaolin clay loadings is shown in SEM images in Figure 2c. The surface of the PBAT/kaolin composite sample in Figure 2c (SEM image) is relatively rough due to the presence of kaolin particles, as compared to the neat PBAT’s smooth surface. In the PBAT, the kaolin clay is distributed evenly in this image. The surface of the PBAT/kaolin composites (inset) in Figure 2c (SEM image) is noticeably rougher than the neat PBAT because kaolin powder has been added. The uniform distribution of kaolin in the PBAT/kaolin composite is further confirmed by the TEM image in Figure 2d. The significant shattering of kaolin during the composite preparation process could be a cause of the significant reduction in structured size.

### 3.2. Thermal Characterization of PBAT/Kaolin Clay Composites

Figure 3a shows TGA thermograms of PBAT and PBAT/kaolin composites. The initial degradation of the PBAT and PBAT/kaolin composite film from 285 to 372 °C can be ascribed to evaporation and the removal of aromatic impurities. With a weight loss of 21.08%, the PBAT matrix (PBAT-0.0) showed the final stage of degradation at 382–460 °C, which was induced by the thermal degradation of -C=O groups in PBAT. The percentage of the Kaolin mineral in the clay samples was interpolated from the TGA peak between 251 °C and 536 °C, usually as a result of dihydroxylation of the kaolinite mineral. Pure kaolin exhibits around 14% weight loss by heating between this temperature range. With a weight loss of 63.2%, PBAT film showed the final stage of degradation from 386 to 441 °C. With the introduction of 1.0, 2.0, 3.0, and 5.0 wt% (PBC-1.0, PBC-2.0, PBC-3.0, and PBC-5.0), kaolin further increased the onset of thermal degradation at 357, 338, 364, and 375 °C, respectively. This could be the result of the C=O groups in PBAT and Si-O-Si groups in kaolin in the composite film producing bonds among themselves and engaging in coordination interaction. When compared to the raw PBAT matrix, the thermal stability of PBC-1.0, PBC-2.0, PBC-3.0, and PBC-5.0 composite film increased by as much as 28–34%. According to our studies, the smaller kaolin clay particles (which have a higher surface area) and more uniform dispersion of these particles in PBAT are responsible for increased thermal degradation temperatures for PBAT/kaolin clay composites. With increasing amounts of particles introduced to the polymer, the residual weight increased, according to the results indicated.

The derivative weight loss curves (DTG), which show the rate at which the materials deteriorate, can be used to study the processes that cause weight loss as a result of heat deterioration more efficiently. For every formulation, these curves appear in Figure 3b. The presence of two peaks in the DTG curves showed that the thermal degradation of the PBAT film occurred in two steps. Kaolin constituents were engaged in the thermal degradation stage, whereas PBAT was the focus of the second. At around 110 °C, a minor peak was observed in the composite formulations. This results from the materials releasing moisture. The low-temperature decomposition (left shoulder of DTG curve) occurred at 300 and 380 °C and could be attributed to the thermal degradation of kaolin, the least thermally stable kaolin component, as well as the decomposition of PBAT. At the 350–450 °C temperature range, the second decomposition step was seen. It was assumed that this area had a connection with the degradation of kaolin. The DTG curve results indicate that the kaolin-added materials followed two-step processes, whereas these samples had a higher decomposition temperature than the control samples exhibited. The amount of char residues that combustion can form on the surface can increase the thermal stability.

### 3.3. Characterization of Mechanical and Barrier Properties

Figure 4a shows the properties of the technically described PBAT and PBAT/kaolin clay composite films that have been studied to evaluate their mechanical characteristics. The tensile strength (TS) and elongation at break (EAB) of kaolin-filled PBAT composites are influenced by filler loading, according to Figure 4a. The TS was observed to significantly increase as the kaolin concentration in the PBAT was increased from 1.0% to 5.0%. With respect to pure PBAT film, the PBAT/kaolin (5.0 wt%) composite film has a higher tensile strength (37.60 MPa) (18.34 MPa). While increased filler concentrations decrease the strength and lower EAB due to the agglomeration of fillers and the filler–matrix interface, it might be the reason for this discontinuity. An elastic material, neat PBAT has a high EAB (570.19%) and a low TS (18.34 MPa). When 1.0 wt% of kaolin is added, the tensile strength increases significantly (19.20 MPa), and the EAB decreases to 523.82%. Increased kaolin concentration at 5.0 wt% reduced EAB to 351.22% and increased PBAT (37.60 MPa). It is probably because the kaolin filler and PBAT exhibit strong interfacial bonding. In terms of stress, a successful interfacial bonding between the matrices and the filler is needed. Moreover, the overall altering behavior of TS increased, and EAB reduced as the amount of kaolin in the films decreased.

In Figure 4b, the WVTR and OTR of PBAT/kaolin composites are presented. H_2_O and O_2_ molecules can pass directly through PBAT without any barriers [44,45]. The OTR was 1080.21 cc/m^2^/day.atm for the PBAT film. After kaolin (5.0 wt%) was mixed on PBAT, it decreased to 311.70 cc/m^2^/day.atm. The value is significantly reduced by the addition of kaolin in different weight %. The minimum value of 783.15 cc/m^2^/day.atm for kaolin was achieved for 2.0 wt%, irrespective of whether the OTR value was still decreased to 960.67 cc/m2/day.atm for 1.0 wt%. The process leading to the rise in permeability is the development of a tortuous path that poses a challenge for gas molecules to flow through the film. Additionally, the orientation and highest shuck (off) level of kaolin in PBAT led to a decrease in OTR. The addition of kaolin to the PBAT matrix significantly decreased the WVTR of the PBAT. According to the measurements, the WVTR values for PBC-0.0, PBC-1.0, PBC-2.0, PBC-3.0, and PBC-5.0 were 152.45, 107.62, 83.17, 60.92, and 43.50 g/m^2^/day, respectively. WVTR decreased with kaolin concentration, and PBAT increased. While the WVTR of pure chitosan is 152.45 g/m^2^/day, that of PBAT/kaolin clay composites varies from 152.45 to 43.50 g/m^2^/day. The decreases in permeability of composites are attributed to the presence of the uniform dispersion of kaolin with increased percentages in the polymer matrix. However, in the PBAT/kaolin composites, these small molecules have migrated through or around the surfaces of impenetrable kaolin, which results in a long and convoluted pathway.

### 3.4. Water Contact Angle Measurement

Hydrophobicity experiments, or WCAs, were performed in order to evaluate the surface characteristics of PBAT/kaolin composites and identify how well the filler content impacted these characteristics. The wettability of a surface is evaluated using the contact angle in order to determine whether it is hydrophilic or hydrophobic. Surfaces with a surface contact angle greater than 90° are referred to as hydrophobic. The contact angle values for PBAT/kaolin clay composites with different kaolin wt% are shown in Figure 5. The contact angle of the PBAT film is 70.5°. In the event that PBAT and the PBAT film were mixed, the contact angle value increased to 93.1°.

This exhibits the hydrophobic character of PBAT. With PBAT containing 1.0 wt% kaolin, the contact angle value was increased to 75.1°. The contact angle value increased to 79.4° and 86.9° for kaolin concentrations of 2.0 and 3.0 wt% in the PBAT matrix, respectively. Kaolin was added, and as a result, the contact angle values increased, indicating the hydrophobic character of PBAT/kaolin clay composite films. Kaolin’s hydrophobic characteristics were produced by the density-functional theory. In comparison to the absorption coefficient on the surface of kaolin, the bond length of hydrogen atoms is high. Water molecule aggregates therefore form on the surface. Hydrophobicity is strongly affected by a variety of additional factors, such as the degree of loading, compatibility, and polymer matrices. Balaji et al. showed that the water contact angle matched similar results [46].

### 3.5. Antimicrobial Activities of PBAT/Kaolin Clay Composites

The inhibition zone against *S. aureus* and *E. coli* shows the results. The PBAT/kaolin composites, in comparison, show a clear antimicrobial zone of inhibition. Figure 6 shows the diameters of the antimicrobial zones in the PBAT/kaolin composites. The (PBC-5.0) composites’ powerful antimicrobial activity against *S. aureus* and *E. coli* suggests that kaolin can increase antimicrobial activity. PBAT/kaolin composites revealed dramatically improved antimicrobial properties against food-borne pathogen microorganisms *E. coli* and *S. aureus* compared to neat PBAT. The zones of inhibition diameter of PBAT/kaolin biocomposites are 10.0, 11.3, 12.6, 14.0, and 16.4 mm against *E. coli* and 10.0, 12.4, 15.1, 18.2, and 19.7 mm against *S. aureus* with loadings of 0.0, 1.0, 2.0, 3.0, and 5.0 wt% kaolin, respectively. The slightly lower antimicrobial zone widths for *S. aureus* relative to those for *E. coli* and *S. aureus* provide additional confirmation that PBAT/Kaolin composite effects are real and significant against *S. aureus* microorganisms. With these materials, PBAT-based biodegradable plastics have a maximum effect and efficient antimicrobial activity.

## 4. Conclusions

We incorporated kaolin clay into PBAT to fabricate biocomposites. The incorporation of kaolin enhanced the mechanical properties of composites made from PBAT. The good interaction of the materials was shown by the FTIR spectra. According to the compatibility between both PBAT and kaolin, kaolin clay is dispersed uniformly in PBAT based on the morphology of the surface. The addition of kaolin clay was found to increase the thermal stability of PBAT composites, which were directly influenced by kaolin clay dispersion and concentration. Kaolin clay was discovered to improve tensile strength and film thickness when it was introduced to PBAT, but it had an opposite result on the film’s elongation at break and barrier properties (OTR and WVTR). The transmission rates to H_2_O and O_2_ composites significantly decreased. With increasing kaolin, the PBAT’s water contact angle improved from 70.5° to 93.1° and improved the hydrophobicity of the composite film. It is interesting to note that the produced composites show significantly improved antimicrobial properties against Gram-positive and Gram-negative bacteria, namely *E. coli* and *S. aureus*. This study suggests that PBAT/kaolin composites offer potential as materials for food packaging to prevent bacterial growth and extend the shelf life of food packages.

## Figures and Tables

**Figure 1 polymers-15-01710-f001:**
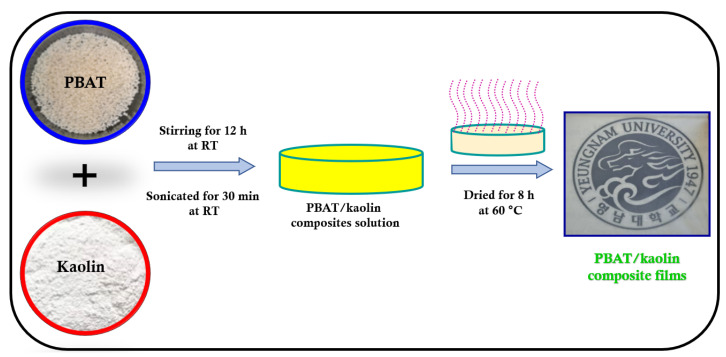
Fabrication diagram for PBAT/kaolin clay biocomposites.

**Figure 2 polymers-15-01710-f002:**
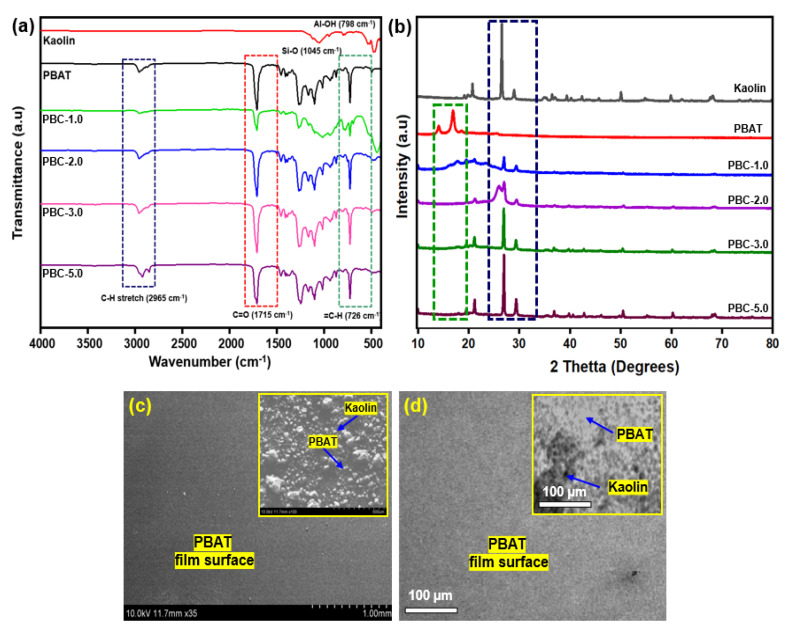
Characterizations of the microstructure and morphology of PBAT/kaolin composites: (**a**) FTIR spectrum; (**b**) XRD; (**c**) SEM image; PBC-5.0 film insets of (**c**); (**d**) TEM image of PBAT and PBAT/kaolin biocomposites (insets (**d**)).

**Figure 3 polymers-15-01710-f003:**
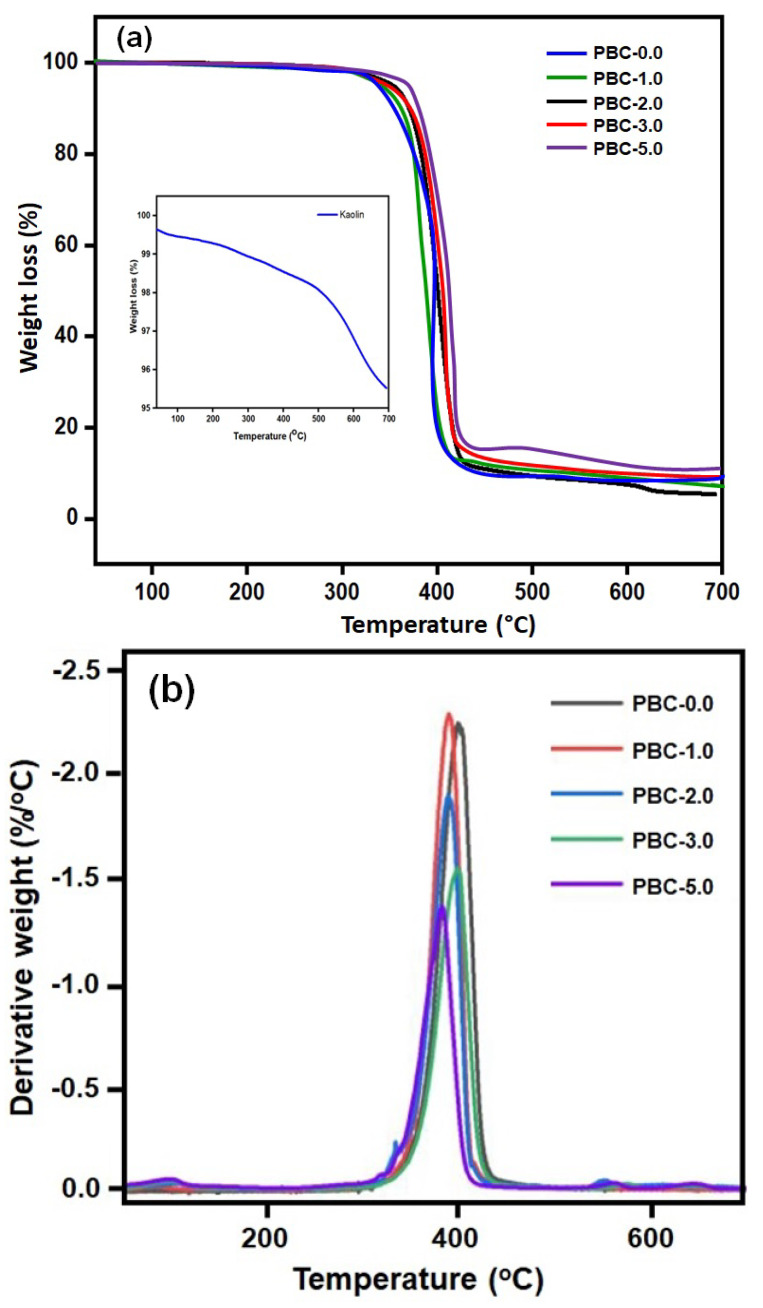
(**a**) TGA curves of PBAT and PBAT/kaolin clay composite samples. (**b**) DTG curves of PBAT and PBAT/kaolin clay composites.

**Figure 4 polymers-15-01710-f004:**
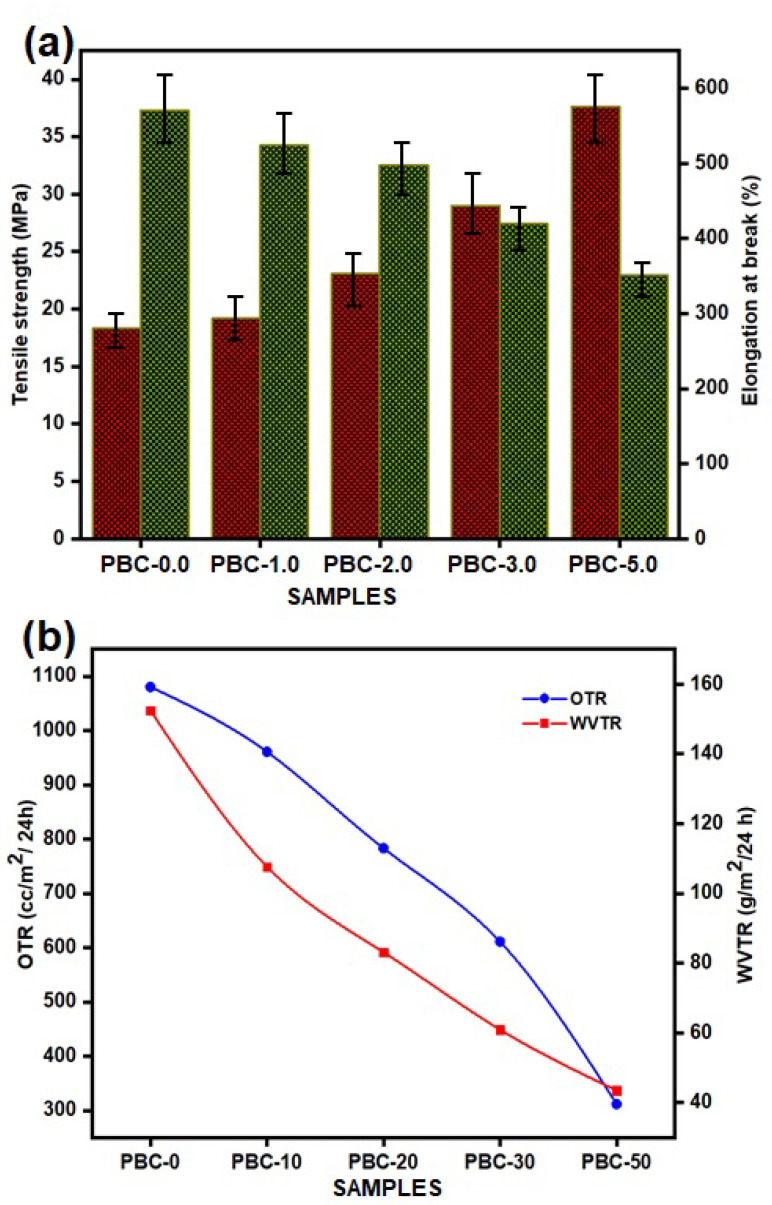
(**a**) Tensile strength and elongation at break of neat PBAT and PBAT/kaolin clay biocomposites; (**b**) OTR and WVTR of PBAT/kaolin clay biocomposites.

**Figure 5 polymers-15-01710-f005:**
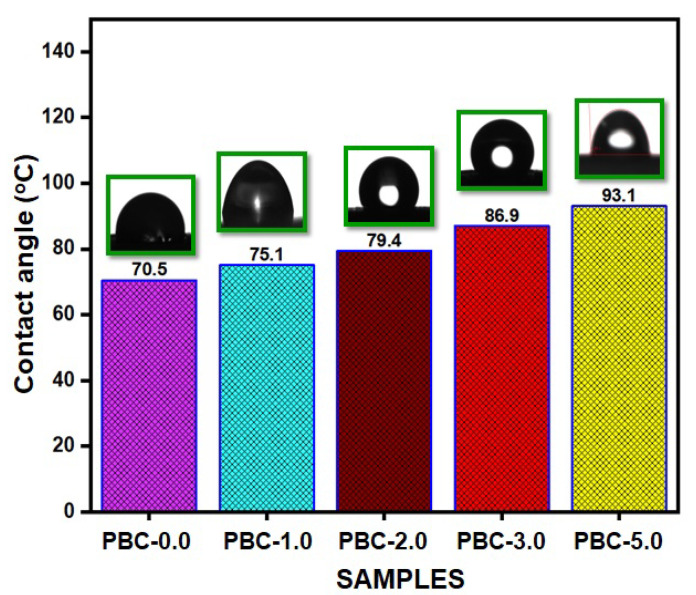
Water contact angle values of PBAT/kaolin biocomposite samples.

**Figure 6 polymers-15-01710-f006:**
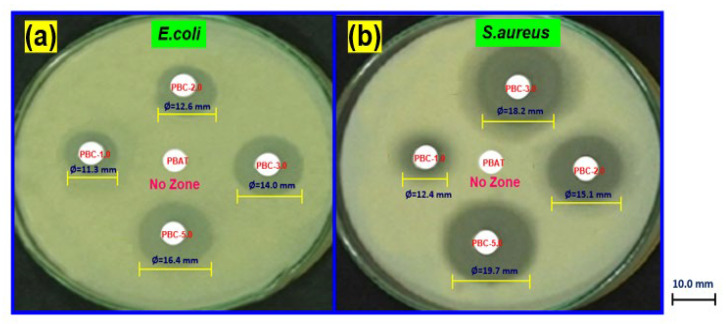
Antimicrobial properties of PBAT/kaolin clay composites with different wt% of kaolin; (**a**) *E. coli*, and (**b**) *S. aureus*.

**Table 1 polymers-15-01710-t001:** Chemical composition of the used kaolin (wt%).

SiO_2_	Al_2_O_3_	Fe_2_O_3_	MgO	K_2_O	CaO	TiO_2_	LOI ^a^
47.85	37.60	0.83	0.17	0.97	0.57	0.74	11.27

^a^ LOI: loss on ignition at 1000 °C.

**Table 2 polymers-15-01710-t002:** Material formulation in blends and composite film preparation.

S. No	Sample Name	PBAT Pellets (wt%)	Kaolin Clay (wt%)
1.	PBC-0.0	100.0	0.0
2.	PBC-1.0	99.0	1.0
3.	PBC-2.0	98.0	2.0
4.	PBC-3.0	97.0	3.0
5.	PBC-5.0	95.0	5.0

## Data Availability

Not applicable.

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
