# Peer review of "Preparation and Antimicrobial Characterization of Poly(butylene adipate-co-terephthalate)/Kaolin Clay Biocomposites"

_polymers, 2023, doi:10.3390/polym15071710_

Round 1

Reviewer 1 Report

In this work, kaolin clay (KO) was blended with poly(butylene adipate-co-terephthalate) (PBAT) to prepare antimicrobial packaging materials using a solution casting method. The materials were characterized by fourier transform infrared spectroscopy (FTIR), X-ray diffraction (XRD), scanning electron microscopy (SEM), and transmission electron microscopy (TEM), Thermogravimetric analysis (TGA), mechanical test, oxygen transmission and water vapour transmission tests, and antimicrobial test.

This work is useful and well designed. The manuscript is well written. Some suggestions to improve the content:

1.      Please give a conclusion for the abstract.

2.  Line92: please do not start a sentence with Fig. 1.

3.  Some methods are not presented in the manuscript, such as oxygen transmission and water vapour transmission tests.

4.  Please add the DTG curves for the films.

5.  Fig. 4C: please add the WCA pictures for all the films.

Author Response

Comments from the reviewers

Reviewer #1:

In this work, kaolin clay (KO) was blended with poly(butylene adipate-co-terephthalate) (PBAT) to prepare antimicrobial packaging materials using a solution casting method. The materials were characterized by fourier transform infrared spectroscopy (FTIR), X-ray diffraction (XRD), scanning electron microscopy (SEM), and transmission electron microscopy (TEM), Thermogravimetric analysis (TGA), mechanical test, oxygen transmission and water vapour transmission tests, and antimicrobial test. This work is useful and well designed. The manuscript is well written. Some suggestions to improve the content:

Response: We are thankful to the reviewer for insightful comments and suggestions and express gratitude for the time spent reviewing our manuscript. We have modified the manuscript in lieu of the reviewer’s comments.

  1. Please give a conclusion for the abstract.

Response: Thank you so much for your kind comments. In the revised paper, issues have been corrected, and the conclusion has been added to an abstract section. (Line 21-23)

  1. Line92: please do not start a sentence with Fig. 1.

Response: The sentence has been changed in the revised manuscript and is highlighted in red in accordance to reviewer suggestions. (Line 94-95)

  1. Some methods are not presented in the manuscript, such as oxygen transmission and water vapour transmission tests.

Response: We included a reference paper and a revised version of this manuscript for the oxygen and water vapor transmission tests. Ref. No: (39 and 40); (Line 160-169).

  1. Please add the DTG curves for the films.

Response: We have added the DTG curves for PBAT and PBAT/kaolin composite film in accordance to reviewer comments. and the text explanation is given to us. (Line 293-307); Fig 3(b).

  1. Fig. 4C: please add the WCA pictures for all the films.

Response: We have added the WCA images of the overall composites to the revised manuscript in response to reviewer suggestions. (Fig. 4(c))

Reviewer 2 Report

The manuscript has investigated the physico-chemical and antimicrobial features of PBAT/kaolin composites with different contents. The topic is interesting and the results are well-discussed. However, it has several problems:

1. L 147,148; According to ASTM standards (D-638), ....  Also, it needs a reference.

2. L 159, please check the citation style.

3. At the end of the materials and methods Section, it is necessary to add the "Statistical analysis" part and check the significant difference between the means.

4. The significant letters should be added to Fig. 4-a and -b. Also, Fig. 4-a should be labeled.

Author Response

Reviewer #2:

The manuscript has investigated the physico-chemical and antimicrobial features of PBAT/kaolin composites with different contents. The topic is interesting and the results are well-discussed. However, it has several problems:

Response: Thank you for pointing out these corrections in the text. These changes have been corrected in the revised manuscript and the text has been carefully examined again. Please read the highlighted parts with red color for details in this revised manuscript.

  1. L 147,148; According to ASTM standards (D-638), .... Also, it needs a reference.

Response: The ASTM standard is added as a reference. (Ref. No. 38).

  1. L 159, please check the citation style.

Response:  In the revised paper, we corrected the citation style.

  1. At the end of the materials and methods Section, it is necessary to add the "Statistical analysis" part and check the significant difference between the means.

Response: In the end of the materials section, we have added a statistical analysis, and we have indicated the significance of the difference depending on reviewer comments. (Line. 189-193)

  1. The significant letters should be added to Fig. 4-a and -b. Also, Fig. 4-a should be labeled.

Response: Thank you very much for your very positive comments. We have corrected them in the revised manuscript.

Round 2

Reviewer 1 Report

The revised manuscript can be accepted.

Reviewer 2 Report

The manuscript is now acceptable.